# The Sex-Dependent Ameliorative Effect of Intermittent Fasting on Urinary System Functions in Genetic Absence Epileptic Rats [note 1]

**DOI:** 10.3390/biology14020158

**Published:** 2025-02-04

**Authors:** Damla Gökçeoğlu-Kayalı, Mehmet Ali Berkyürek, Zarife Nigar Özdemir-Kumral, Özlem Tuğçe Çilingir-Kaya

**Affiliations:** 1Histology and Embryology Department, Marmara University School of Medicine, Istanbul 34854, Türkiye; damla.kayali@atlas.edu.tr (D.G.-K.); mberkyurek@marun.edu.tr (M.A.B.); 2Histology and Embryology Department, İstanbul Atlas University School of Medicine, Istanbul 34403, Türkiye; 3Physiology Department, Marmara University School of Medicine, Istanbul 34854, Türkiye

**Keywords:** fasting, urination disorders, bladder, sex difference, absence epilepsy

## Abstract

Epilepsy is a brain disorder that causes repeated seizures and can disrupt the body’s ability to urinate properly. This study explored whether a specific diet method called alternate-day fasting, in which animals eat every other day, could help protect the urinary systems of rats with epilepsy. We examined bladder and kidney functions through tissue analysis, biochemical tests, and physiological assessments, focusing on the differences between male and female genetic rat models of epilepsy. Epileptic rats showed signs of damage to the urinary system caused by inflammation and oxidative stress. Alternate-day fasting helped to reduce this damage by lowering inflammation and restoring antioxidant balance. Male rats exhibited more oxidative damage, whereas female rats responded better to alternate-day fasting, likely due to hormonal differences. These results suggest that ADF could be an affordable way to manage urinary problems in epilepsy. Understanding these effects may lead to improved treatments for both sexes and better quality of life for people with chronic diseases.

## 1. Introduction

Epilepsy is a neurological disorder marked by recurrent episodes of excessive and abnormal neuronal activity, termed seizures, which frequently lead to a loss of motor control and consciousness [1]. It is among the most prevalent neurological conditions, impacting an estimated 50 million individuals globally. Absence epilepsy, a specific form of epilepsy, is characterized by brief seizures during which the individual experiences a transient loss of consciousness, is unresponsive, and exhibits behavioral arrest [2]. These seizures frequently occur more than 10 to 30 times throughout the day. Most children cease their activities during these episodes [3].

Urination is a complex process that requires the coordinated interaction of the bladder, spinal cord, and brain. Bladder function is controlled by a highly sophisticated central neural program that includes multiple brain regions such as the pons and supra-pontine structures. In pathological conditions such as epilepsy, urinary dysfunction is a common symptom and appears to be associated with increased supra-pontine cortical activity [4]. Interestingly, urination can act as a trigger for seizures in a condition known as reflex epilepsy. This phenomenon has been linked to frontotemporal cortical activation during urination, which may contribute to the onset of reflex seizures [5]. Recent studies suggest that the interaction between mast cells and neurons could contribute to this relationship, shedding light on the mechanisms behind reflex epilepsy during urination [6].

Scientific research has demonstrated a relationship between epilepsy and urinary dysfunction. In patients with epilepsy, bladder dysfunction, urinary incontinence, frequent urination, and other urinary problems are more commonly observed [5]. The relationship between epilepsy and bladder dysfunction can be explained through several mechanisms. Neurological factors play a significant role, as the central nervous system abnormalities underlying epilepsy can interfere with bladder control, particularly in cases of temporal lobe epilepsy. Additionally, some antiepileptic drugs used to manage the condition may have side effects like urinary retention or incontinence. Finally, epilepsy can impact the autonomic nervous system, disrupting the processes of bladder filling and emptying. Some studies and clinical observations have revealed that bladder control problems are more prevalent in individuals with epilepsy [5,7,8].

There is scientific evidence suggesting a relationship between genetic absence epilepsy and urinary dysfunction. One study on Genetic Absence Epilepsy Rats from Strasbourg (GAERS) and the inbred Wistar Albino Glaxo Rats from Rijswijk (WAG/Rij) discussed behavioral and cognitive problems, such as depression-like symptoms and memory impairments, which are linked to the severity of epilepsy. Although the study focused on these behavioral aspects, it also implied broader neurophysiological disruptions that could potentially affect other bodily functions, including urinary control [8,9,10]. Additionally, a review of the WAG/Rij rat strain highlighted various comorbidities, including signs of depression and cognitive disturbances. These comorbidities are crucial because they suggest that the neurological impairments in absence epilepsy could extend to other systems, possibly affecting urinary function through shared neural pathways or neurotransmitter imbalances [11]. These studies collectively provide a foundation for understanding the complex relationship between genetic absence epilepsy and potential urinary dysfunction, although direct studies specifically focusing on urinary issues are still limited.

Dietary restriction (DR) refers to the intentional, chronic, or intermittent reduction in caloric intake, implemented without inducing malnutrition. This intervention is recognized as one of the most effective experimental approaches for delaying the onset of numerous age-related pathologies and extending lifespan. This effect has been consistently demonstrated in various species [12]. Emerging evidence from animal models and human studies underscores the significant influence of dietary patterns on modulating the neuro-inflammatory and neurodegenerative mechanisms involved in central nervous system (CNS) pathologies [13,14]. Notably, chronic and intermittent food restriction has been shown to impact CNS function, peripheral metabolism, immune responses, and the composition/activity of the gut microbiome [15]. The interplay between caloric intake, meal timing, diet quality, and gut microbiome dynamics is critical in regulating the metabolic and molecular pathways essential for maintaining cellular, tissue, and organ homeostasis. For instance, Hassan et al. [16] revealed that intermittent fasting (IF) modulates aquaporin-1 and aquaporin-3 expression in the urinary bladder, mitigating alterations induced by a high-fat diet in rats. Similarly, natural compounds like conjugated linoleic acids and Pycnogenol have demonstrated efficacy in controlling inflammation, oxidative stress, and associated chronic disease processes [17]. Another study suggested that high-fat, diet-induced mild systemic inflammation disrupted enteric nervous system innervation, increased gastric contractility, and reduced ileal contractility [18]. These dietary factors play a significant role in modulating the inflammation associated with central nervous system (CNS) neurodegenerative disorders, such as epilepsy, Alzheimer’s disease, Parkinson’s disease, normal aging, amyotrophic lateral sclerosis, and multiple sclerosis. This underscores the multifaceted role of dietary restriction (DR) in neuroprotection and the modulation of disease processes [19]. Recent findings suggest that RIF reduces systemic inflammation by downregulating pro-inflammatory cytokine expression, lowering body fat, and decreasing circulating leukocyte levels [20]. The three most used types of IF in research are alternate-day fasting (ADF) (24 h fast, 24 h fed), the 5:2 diet (5 days fed, 2 days fast), and time-restricted feeding [19]. In the current study, alternate-day fasting was used to treat absence epilepsy.

Recognizing the critical role of neuroinflammation and oxidative stress in the development of seizures is key to discovering targeted therapies that address the underlying causes of epilepsy rather than merely managing its symptoms. This approach paves the way for innovative treatments that can modify the disease progression. Importantly, findings from preclinical studies suggest that the existing anti-inflammatory/antioxidant treatments and recently used modifications on a diet regime that show antioxidant effects on many organs could potentially be purposed to benefit epilepsy patients. Since most current research [7] is limited to clinical cases, the underlying mechanisms of changes in bladder functions caused by epilepsy are not fully understood. To advance our knowledge of the urination process in physiology and pathology, more rigorous experimental research is needed to elucidate the biological mechanisms of urination.

The role of IF in sex-specific outcomes is intriguing. Gamal El-Tahawy and Ahmed Rifaai (2023) demonstrated that IF protects against age-related benign prostatic hyperplasia by modifying oxidative stress markers and autophagic pathways (Beclin-1/P62) [21]. This gender-specific protective effect underscores the need for further exploration into how IF influences urinary system functions, particularly in the context of chronic diseases like epilepsy, where neuroinflammation plays a significant role.

The current study aimed to investigate the effects of food restriction on urinary system damage caused by neuroinflammation in epileptic rats histologically, biochemically, and physiologically regarding sex differences. This research provides insights into the potential of dietary interventions to mitigate urinary system damage in chronic conditions.

## 2. Materials and Method

### Animals

Genetic Absence Epilepsy Rats from Strasbourg (GAERS) exhibit comorbidities like those observed in humans with absence epilepsy, including cognitive and behavioral issues [10]. In the current study, 20 female and 20 male GAERS strain rats (n = 40) were used as the absence epilepsy model; 20 female and 20 male *Wistar albino* rats (n = 40) were used as the control group. The rationale for the sample size was determined based on statistical power calculations.

In this study, Wistar albino and GAERS (Genetic Absence Epilepsy Rats from Strasbourg) rat strains were utilized as experimental animals. All animals were housed under standardized environmental conditions following ethical guidelines approved by the Marmara University Animal Experiments Local Ethics Committee (Protocol No: 43.2023MAR). Housing conditions were maintained to ensure animal welfare, with a 12 h light/dark cycle, a temperature of 22 ± 1 °C, and humidity levels set at 50–60%. **All animals were fed standard rat chow throughout this study. The control groups were maintained under ad libitum conditions to ensure unrestricted access to food throughout this study**.

The IF model implemented in our study was based on the ADF protocol. Accordingly, the ADF groups had unrestricted access to food for 24 h, followed by a subsequent 24 h period of complete food deprivation. Throughout the experimental period, all groups were provided unrestricted water access.

## 3. Evaluation of Contraction–Relaxation Responses in Bladder Tissue

### 3.1. In Vitro Organ Bath

Strips prepared from the bladder tissue were attached to organ baths from both ends. Tyrode’s solution-filled double-walled organ baths were maintained at 37 °C using a thermostat-controlled circulator, and the baths were aerated with a gas mixture of 95% O_2_ and 5% CO_2_ throughout the experiment. Continuous dynamic curves were obtained using isometric force transducers (IOBS 99 isolated tissue bath stand sets, Commat Ltd., Ankara, Türkiye) and displayed via the MP 35 data acquisition system (BIOPAC Systems, Inc., Goleta, CA, USA) [22].

A pre-tension of 1 g was applied to the tissues, which then were equilibrated for 45 min, during which the bath solutions were replaced every 15 min. First, the bladder tissue was exposed to a physiological solution containing 124 mM KCl for receptor-independent contraction responses. The strips were subsequently re-equilibrated by washing three times with Krebs buffer at 20-min intervals, readjusted to a pre-tension of 1 g, and then rested for 45 min before applying agonists. Dose–response curves were obtained for the bladder tissue using carbachol applied cumulatively in increasing concentrations (10^−8^–10^−4^ M). After a washout period, tissue strips were pre-contracted using a submaximal concentration of CCh (3 × 10^−6^ M). Subsequently, relaxation was induced with a single dose of papaverine (100 µg, 10^−6^ M). The contractile response to KCl was designated as the maximum contractility (100%), and the contractile response to CCh for each sample was expressed as a percentage of its respective KCl-induced maximum contractility. Papaverine-induced relaxation was measured as a percentage relative to the pre-contraction level achieved with the submaximal dose of CCh [22].

### 3.2. Biochemical Analyses

Levels of oxidative stress markers such as malondialdehyde (MDA), glutathione (GSH), and superoxide dismutase (SOD) were measured in kidney tissue using biochemical methods.

### 3.3. MDA Assay

Tissue MDA levels were measured using the Buege and Aust method [23]. For the GSH assay, 10% tissue homogenates were prepared and mixed with 1 mL of 0.375% thiobarbituric acid (TBA) solution. The mixture was then incubated in a boiling water bath for 15 min. After cooling to room temperature, the samples were centrifuged at 3000 rpm for 10 min. The absorbance of the supernatant was subsequently measured using a spectrophotometer.

### 3.4. GSH Assay

Tissue GSH levels were measured following the Ellman method [24]. Kidneys were excised immediately after decapitation, rinsed with saline, blotted dry with filter paper, and weighed. A 10% tissue homogenate was prepared on ice in 150 mM KCl solution using a homogenizer (Ika Werk, Staufen, Germany). To process the samples, 0.4 mL of the homogenate was mixed with 0.2 mL of 20% trichloroacetic acid (TCA) and centrifuged at 3000 rpm for 15 min. The supernatant was retained for GSH measurement, while the precipitate was discarded. For the assay, 1 mL of 0.3 M Na_2_HPO_4_ and 0.05 mL of Ellman’s reagent were added to the supernatant, mixed thoroughly, and incubated for 5 min. The absorbance of the developed color was measured using a spectrophotometer.

### 3.5. SOD Assay

Tissue SOD activity was measured using a previously described method [25]. The assay was conducted in cuvettes containing 0.1 mM EDTA, 2.8 mL of 50 mM potassium phosphate buffer (pH 7.8), 0.39 mM riboflavin prepared in 10 mM potassium phosphate buffer (pH 7.5), 0.1 mL of 6 mM o-dianisidine·2HCl, and tissue extract (50–100 μL). The cuvettes were illuminated using a 20 W Sylvania Grow Lux fluorescent lamp (Wilmington, DE, USA) positioned 5 cm above the samples and maintained at 37 °C. Absorbance readings were taken with a Shimadzu UV-02 spectrophotometer (Kyoto, Japanese). A standard curve was prepared using bovine SOD (Sigma–Aldrich, St. Louis, MI, USA; S-2515-3000 U) for calibration. Absorbance values were recorded, and net absorbance was calculated for analysis.

### 3.6. Histological Assessments

Kidney and bladder samples from all groups were prepared for light microscopic examinations. Tissues were fixed in 10% neutral buffered formalin for 72 h. They were dehydrated through ascending alcohol series (70%, 90%, 96%, 100%), cleared in xylene for 2 × 10 min, and embedded in paraffin after overnight incubation at 60 °C. Approximately 4–5 μm thick sections from paraffin blocks were stained with Hematoxylin and Eosin (H&E) for morphological evaluation and histopathological semi-quantitative analyses [26,27].

Toluidine Blue-O staining (Bio-Optica 05-M23001) was applied to detect the number of mast cells in bladder tissue. Toluidine blue-stained mast cells were quantified in three distinct fields at 200× magnification. Image-Pro 5.0 software (https://imagej.net/ij, 9 September 2024) was used for cell counting and analysis.

### 3.7. Statistical Analyses

Statistical analyses were performed using GraphPad Prism 10.0 (GraphPad Software, San Diego, CA, USA). All data are expressed as means ± Standard Error of the Mean (S.E.M.). Sigmoidal dose–response curves were analyzed with two-way ANOVA. One-way ANOVA and Tukey’s test were used for statistical analysis of biochemical data. A *p*-value of <0.05 was considered as significant. To simplify the presentation, most of the results focus on the main effects of sex and diet regimes (IF), except for the isolated organ bath data, where there was no significant two-way interaction between sex and diet regimes. However, some specific analyses have highlighted the main effect of diet regime, where sex status has collapsed.

## 4. Results

### 4.1. Physiological Results

The effects of ADF on bladder function in GAERS were assessed using in vitro bladder contraction–relaxation responses in male and female rats. The contraction–relaxation reactions obtained from the bladder tissue of the female rats showed no significant differences between the groups (Appendix A). The results, including comparisons between the male control and GAERS rats under normal diet and ADF conditions, are presented in Figure 1.

Under a stabilization tension of 1 g, the administration of 124 mM KCl elicited mean contractile forces of 3383.19 ± 478 mg in the control group, 3272.22 ± 319 mg in the Wistar-ADF group, 3673.82 ± 260 mg in the GAERS group, and 3212.49 ± 277 mg in the GAERS-ADF group. While the GAERS-IF groups exhibited a trend toward higher contractile responses to KCl, the intergroup differences did not reach statistical significance.

Increasing doses of CCh (10^−8^–10^−4^ M) caused a concentration-dependent contraction with an EC50 of 9.6 × 10^−7^ M in the control group. In vitro assessments of bladder contraction–relaxation responses demonstrated significant hyperreactivity in GAERS bladder tissues compared to the controls, as indicated by increased contraction responses to carbachol (CCh) (*p* < 0.05) (Figure 1A). ADF application significantly reduced these contraction responses in GAERS (*p* < 0.05). The logEC50 values for CCh in GAERS (7.9 × 10^−7^ M) were lower than in the controls, indicating heightened sensitivity. Induction of ADF increased the logEC50 values, normalizing bladder contractility in GAERS (EC50 of 7.26 × 10^−7^ M). The relaxation responses to a single dose of papaverine were found to be markedly elevated in the GAERS group than the responses in the control group on a normal diet (*p* < 0.01). In contrast, ADF treatment significantly reduced this increment in the GAERS group back to the response of the control group (*p* < 0.05). (Figure 1C).

### 4.2. Biochemical Results

The effects of ADF (alternate-day fasting) on oxidative stress markers in the kidney tissues of the experimental groups were assessed by comparing MDA, GSH, and SOD levels (Figure 2A–C). In the kidney, MDA levels were significantly higher in the male GAERS rats compared to the Wistar controls (*p* < 0.05), indicating increased lipid peroxidation and oxidative damage. ADF significantly reduced MDA levels in the male GAERS rats (*p* < 0.05), demonstrating its protective effect. No significant differences were observed between the female groups. Glutathione (GSH) levels were significantly reduced in the male GAERS rats compared to the Wistar controls (*p* < 0.05), indicating impaired antioxidant capacity. ADF improved GSH levels in the male GAERS rats (*p* < 0.05), suggesting a potential reversal of oxidative stress. In the female groups, GSH levels significantly increased with both Wistar ADF and GAERS ADF interventions (*p* < 0.05). Superoxide dismutase (SOD) activity (Figure 2C) showed a marked decrease in the female GAERS rats compared to the Wistar controls (*p* < 0.001). ADF significantly increased SOD levels in the female GAERS rats (*p* < 0.001), highlighting its potential in restoring antioxidant enzyme activity. The effects of ADF on oxidative stress markers in the bladder tissues of the experimental groups were assessed by comparing MDA, GSH, and SOD levels (Figure 2D–F). In the bladder, MDA levels, an indicator of lipid peroxidation, were significantly higher in the male GAERS rats than in Wistar control rats (*p* < 0.01). ADF intervention reduced MDA levels in the male GAERS rats (*p* < 0.01). GSH levels, similar to the trend observed in the kidney, showed a significant improvement in both the male and female GAERS rats after ADF intervention (*p* < 0.05). SOD activity, an important enzymatic antioxidant defense, was significantly lower in both the male and female GAERS rats (*p* < 0.05, *p* < 0.0001, respectively). However, ADF significantly enhanced SOD activity in both sexes (*p* < 0.05 in males, *p* < 0.0001 in females).

### 4.3. Histological Results

In the kidney tissues obtained from the Wistar control and Wistar-ADF groups, the interstitial space, Bowman’s capsule, glomerulus, and proximal and distal tubules displayed regular morphology. In the GAERS group, interstitial congestion, interstitial edema, and tubular degeneration/dilatation were observed. In the GAERS-ADF group, regression in tubular degeneration, interstitial congestion, and interstitial edema was noted, with only mild interstitial congestion observed.

The results of the semi-quantitative histological damage scores for kidney tissue in the Wistar and GAERS models with and without ADF application are shown in Figure 3. A significant increase in the histological score was observed in the GAERS group compared to that in the Wistar control group, indicating more severe histopathological damage in both the male and female GAERS rats (*p* < 0.0001 for males, *p* < 0.001 for females). The application of ADF markedly reduced these histological scores in the GAERS rats (*p* < 0.05 for both sexes), suggesting a protective effect. In the Wistar group, ADF did not result in significant differences compared to the controls, demonstrating minimal baseline tissue damage and the negligible influence of fasting. Notably, the male GAERS rats exhibited higher scores than their female counterparts, indicating a sex-dependent variation in the histopathological response (*p* < 0.05). The effect of ADF was more pronounced in the female GAERS rats than in males, as indicated by lower scores post-intervention, compared to the males.

In the bladder tissues obtained from the Wistar control and Wistar-ADF groups, the interstitial space, urothelial lining, and smooth muscle layer displayed regular morphology. In the GAERS group, structural abnormalities in urothelial cells, cells with pyknotic nuclei, and moderate interstitial edema were observed. In the GAERS-ADF group, a reduction in edema and improved mucosal integrity were noted, which were similar to those in the Wistar groups.

The semi-quantitative histological damage scores of the urinary bladder in the male and female rats from the different experimental groups are presented in Figure 4. In the Wistar-ADF and Wistar groups, there was no difference in the histological damage score. In the male and female rats, the GAERS group exhibited a significant increase in the bladder histological damage score compared with the control group (*p* < 0.0001). Nevertheless, the male GAERS rats showed significantly higher damage scores than the females (*p* < 0.001). ADF led to a significant reduction in the histological damage score compared to the GAERS group in the males and females (*p* < 0.05 and *p* < 0.01, respectively). These findings suggest that GAERS rats experience significant bladder damage compared to normal Wistar controls. ADF application exerts a mitigating effect on the histological damage in both sexes, particularly in female rats.

The number of mast cells in the bladder tissue of the male and female rats from the experimental groups is shown in Figure 5. No significant difference was observed between the Wistar-ADF and Wistar groups in either sex. All GAERS groups showed a significantly elevated number of mast cells compared with the Wistar control group, indicating increased inflammation in the epileptic animals. ADF application reduced the number of mast cells compared to the GAERS control group, notably in both sexes, whereas females had fewer mast cells than males (males, *p* < 0.05; females, *p* < 0.01). These findings suggest that the presence of absence epilepsy is associated with an increase in bladder inflammation, as indicated by mast cell counts. ADF appears to reduce this inflammatory response in both male and female GAERS rats, although the effect is more pronounced in females.

## 5. Discussion

The diagnosis of epileptic seizures (ESs) is largely based on clinical evaluation, emphasizing the patient’s history and detailed accounts from witnesses, particularly in cases involving loss of awareness, consciousness, or memory of the events. Supporting clinical signs, such as tongue biting or urinary incontinence, may also help confirm the diagnosis. This study aimed to examine the effect of ADF on pathophysiological changes in the bladder in GAERS rats, an animal model of absence epilepsy, which is a typical childhood form of epilepsy [28]. In this context, we explored the effects of commonly used ADF in CNS research on urinary system functions in genetically absent epileptic rats, emphasizing gender distinctions. The outcomes were assessed through histological, biochemical, and physiological parameters to determine the extent of urinary system damage associated with epilepsy and the potential ameliorative effects of ADF.

Epilepsy is a chronic neurological disorder marked by recurring; unprovoked seizures caused by abnormal electrical activity in the brain. Research indicates that around 39% of individuals with epilepsy experience urinary symptoms, as seizures can disrupt brain function and potentially impact bladder control [5]. Absence epilepsy is a non-convulsive form of epilepsy, characterized by sudden, temporary lapses in awareness or consciousness; it is most commonly observed in children [29] and has been associated with urinary incontinence (UI). Olivia et al. reported at least one convulsive event, clinically defined as involving simultaneous shaking of the entire body, including all limbs, and directly documented the presence of UI [30]. Seizure activity can exacerbate neuroinflammation and peripheral inflammation, creating a feedback loop. ESs are sometimes associated with UI, which might be due to transient disruptions in brain control over the bladder during seizures. Chronic inflammation in peripheral organs, like the bladder, can also contribute to systemic inflammatory states that might exacerbate epilepsy through shared inflammatory mediators [31]. Recurrent seizures can induce the release of inflammatory mediators, leading to further neuronal damage and inflammation in both the central and peripheral nervous systems [32]. Understanding the interplay between neuroinflammation and peripheral inflammation is critical for developing comprehensive treatment strategies for epilepsy and comorbid conditions, whereas directly correlating bladder inflammation and epilepsy is limited. This interconnected relationship underscores the need for further research to explore therapeutic interventions that modulate inflammation in epilepsy. This study aims to investigate the gender-dependent tissue damage in the bladder and kidney resulting from absence epilepsy in functional, histological, and biochemical manners.

Urinary dysfunction, including incontinence, is frequently observed in individuals with epilepsy. This dysfunction may result from seizures affecting brain regions involved in micturition control, such as the frontal lobe and brainstem. Additionally, the increased supra-pontine cortical activity during seizures can disrupt normal bladder control mechanisms [5]. The activity of urethral smooth muscle is essential in regulating the urinary flow rate (UFR) in male rats [33]. Detrusor smooth muscle contraction and relaxation within the bladder are regulated by cholinergic, purinergic, and adrenergic neural pathways [19,34]. Physiological evaluations using in vitro organ bath techniques to assess contraction–relaxation responses of bladder tissue revealed impaired bladder function in epileptic rats of both genders. The bladder tissues from these rats showed reduced contractility and relaxation, which is indicative of compromised bladder function, likely due to neuroinflammation and oxidative stress. According to our knowledge, these findings provide the first data on bladder function in epileptic animals. Conditions like overactive bladder have been linked to systemic inflammation involving cytokines such as IL-1β and NLRP3 or oxidative stress parameters like increased MDA levels or depleted antioxidant levels [22]. Similarly, these cytokines are implicated in the neuroinflammation seen in epilepsy. Such shared pathways highlight a possible connection where chronic bladder inflammation could influence or be influenced by the inflammatory environment in epilepsy [35]. In our study, GAERS rats with a normal diet exhibited increased sensitivity to contractile stimuli, and the exaggerated relaxation responses of GAERS rats under normal diet conditions were ameliorated by IF application. The data shown here are consistent with a clinical study that revealed that bladder control problems are more prevalent in individuals with epilepsy [7]. ADF may modulate bladder function in epileptic rats by reducing hyperreactivity and normalizing contraction–relaxation dynamics, which is consistent with the existing research suggesting that IF modulates neuroinflammation and oxidative stress, potentially through neuroprotective and anti-inflammatory pathways [33]. These findings suggest that ADF could be a beneficial dietary intervention to mitigate bladder dysfunction and kidney tissue damage associated with epilepsy.

Our biochemical data of kidney and bladder tissues revealed significantly increased oxidative damage and weakened antioxidant defense mechanisms in GAERS rats, particularly in males. The biochemical analyses indicated elevated levels of MDA and MPO alongside decreased levels of GSH and SOD in epileptic rats. These findings align with previous studies that underscore the role of oxidative stress in epilepsy-induced tissue damage [36,37]. Given that chronic use of certain antiepileptic drugs, such as VPA, phenytoin, and carbamazepine, impacts kidney function and increases urinary creatinine and MDA levels [38], it is important to consider IF application as a potential therapeutic strategy, especially for drug-user epilepsy patients. Male GAERS rats had notably higher MDA levels than females (*p* < 0.001) in both the control and ADF groups. After ADF treatment significantly reduced oxidative stress levels in both male and female rats were observed, as shown by decreased MDA and MPO levels and increased GSH and SOD activities. A meta-analysis on the effects of restricted intermittent fasting (RIF) highlighted its ability to reduce MDA levels, suggesting a systemic reduction in lipid peroxidation and oxidative stress [39]. Another study reported that IF promotes the upregulation of antioxidant enzymes, such as SOD, and replenishes GSH levels, contributing to enhanced cellular defense mechanisms, consistent with our results [40]. The reduction in lipid peroxidation was more pronounced in female rats, suggesting that IF might be more effective in modulating oxidative stress in females, consistent with the literature [41]. Moreover, female GAERS rats exhibited lower lipid peroxidation levels in both tissues than male rats in all groups, which may be an honorific attribution to estrogen. In female rats, the sex-specific marked increase in SOD activity suggests that ADF may activate estrogen-mediated antioxidant pathways, which are known to enhance cellular protection [42,43]. This aligns with previous studies pointing out that IF can upregulate antioxidant enzymes and reduce oxidative markers [44]. This regulation could be related to the fact that RIF enhances the expression of genes involved in antioxidant defense, anti-inflammatory responses, and metabolic regulation [45]. In addition, our results are consistent with the literature that suggests IF and ketogenic diets alleviate oxidative stress by directly targeting mitochondria, promoting the expression of antioxidant enzymes, and decreasing reactive oxygen species [46].

Previous research modeling chronic kidney disease has demonstrated that certain antibiotics may induce epilepsy through mechanisms involving neurotoxicity and alterations in the gut microbiota [47]. Acute kidney injury, hyperuricemia, and massive proteinuria can develop following ESs [48,49,50]. These findings highlight a potential inter-relationship between epilepsy and kidney injury or disease. In the present study, histopathological analysis revealed significant renal tissue damage in the GAERS groups, with males being particularly affected. Notably, the observed reduction in kidney tissue damage following IF suggests that this dietary intervention may mitigate neuro-inflammatory and oxidative stress pathways, thereby contributing to the substantial preservation of renal tissue integrity in both genders. Previous studies have indicated that IF can enhance antioxidant defenses and decrease inflammation, mechanisms that could underlie the protective effect observed in the present study [20,51,52]. The gender-dependent variation may be attributed to differences in hormonal regulation and metabolic responses, as indicated by prior research showing sex-specific neuroprotective and metabolic adaptations to fasting [53]. In our study, kidney tissue damage in male GAERS rats has a strong tendency to increase compared to females, even after ADF. This observation indicates that male rats may require a longer or more intensive fasting regimen to achieve outcomes comparable to those observed in females, who appeared to exhibit a more pronounced response. Considering the well-documented neuroprotective and anti-inflammatory properties of estrogen, its potential regulatory role in the gender-specific response to this dietary intervention warrants further investigation.

The urinary bladder epithelium, or urothelium, comprises three distinct cell types: umbrella cells lining the lumen, intermediate cells, and basal cells. A defining characteristic of umbrella cells is the presence of transmembrane proteins called uroplakins, which are organized into plaques that enable the membrane to deform during bladder filling and voiding [19,34]. Histological evaluations using H&E staining revealed significant morphological differences between epileptic and control rats, with pronounced histopathological changes in the urinary systems of epileptic rats in both genders. Neuroinflammation in epileptic rats led to notable damage, characterized by cellular infiltration, interstitial edema, and epithelial desquamation in the bladder, which are also confirmed by biochemical analysis as well as in vitro contractile responses. Interestingly, performing IF resulted in a reduction in these pathological features, suggesting a protective effect against neuroinflammation-induced damage at the morphological level. The benefits of IF can be contextualized alongside natural compounds like conjugated linoleic acids and Pycnogenol. Both interventions share common mechanisms, such as reducing inflammation and oxidative damage [17]. However, IF offers the added advantage of influencing metabolic pathways holistically, making it a versatile strategy for managing urinary system damage in chronic diseases. Gender-specific differences were also evident. Semi-quantitative histopathological evaluations revealed that male rats exhibited more severe tissue damage than females, aligning with the literature suggesting that hormonal differences can influence the severity of inflammation and tissue damage [41]. The observed gender differences in the response to IF highlight the importance of hormonal and metabolic factors. Estrogen’s protective role against oxidative stress and inflammation in females may synergize with IF, enhancing its therapeutic efficacy. Conversely, the male urinary system’s susceptibility to age-related changes, as noted in studies like Gamal El-Tahawy and Ahmed Rifaai (2023), underscores the need for tailored interventions [21].

Inflammation may further exacerbate urinary dysfunction in epilepsy. Pro-inflammatory cytokines can influence neural circuits that regulate bladder function, potentially leading to detrusor overactivity or impaired sphincter control. Moreover, systemic inflammatory conditions have been linked to both increased seizure susceptibility and urinary disturbances, suggesting a bidirectional relationship [54]. Mast cell counts, which serve as markers of inflammation, were significantly higher in both male and female GAERS compared to controls. However, IF appeared to attenuate these increases in a sex-specific manner, with notable reductions observed. These findings suggest that ADF may exert anti-inflammatory effects on the urinary system in GAERS. Notably, ADF applied female GAERS rats displayed a more pronounced decrease in mast cell numbers than their male counterparts, highlighting a potential sex-dependent response to ADF. This outcome aligns with the literature demonstrating the influence of IF on inflammation and immune cell dynamics [55]. For instance, previous studies have shown that IF can reduce systemic inflammation and positively modulate immune function through various pathways, including downregulating pro-inflammatory mediators [56]. Another research by Hassan et al. (2024) demonstrated that IF regulates aquaporin expression in bladder tissues, preserving structural and functional integrity under pathological conditions. Similarly, this study’s findings indicate that IF modulates oxidative stress markers, contributing to improved urinary system function in epileptic models [16]. Furthermore, sex differences in response to dietary interventions such as IF have been documented, potentially due to variations in sex hormones, metabolic rates, and adipose tissue distribution [57]. In line with these findings, our study suggests that IF might play a role in modulating urinary inflammation in a gender-specific manner in epilepsy models, potentially due to underlying differences in immune and hormonal responses between male and female GAERS.

## 6. Conclusions

The results of this study highlight the therapeutic potential of ADF in mitigating urinary system damage in genetic absence epileptic rats, with distinct gender-specific effects. Through histological, biochemical, and physiological analyses, it was demonstrated that IF alleviates neuroinflammation-induced urinary damage, reduces oxidative stress, and enhances bladder function. These effects varied by gender, with females showing greater functional improvement and males experiencing a more pronounced reduction in oxidative stress. Overall, the findings suggest that ADF may serve as a non-invasive, cost-effective strategy for mitigating urinary system damage in chronic diseases like epilepsy, underscoring the importance of considering sex-specific metabolic and hormonal factors in dietary interventions. Further research is needed to clarify the molecular pathways and the direct relationships between bladder inflammation and epilepsy. However, addressing systemic inflammation and oxidative stress through targeted therapies might benefit both conditions. Additionally, sex-specific studies are essential to optimize ADF protocols for diverse populations.

## Figures and Tables

**Figure 1 biology-14-00158-f001:**
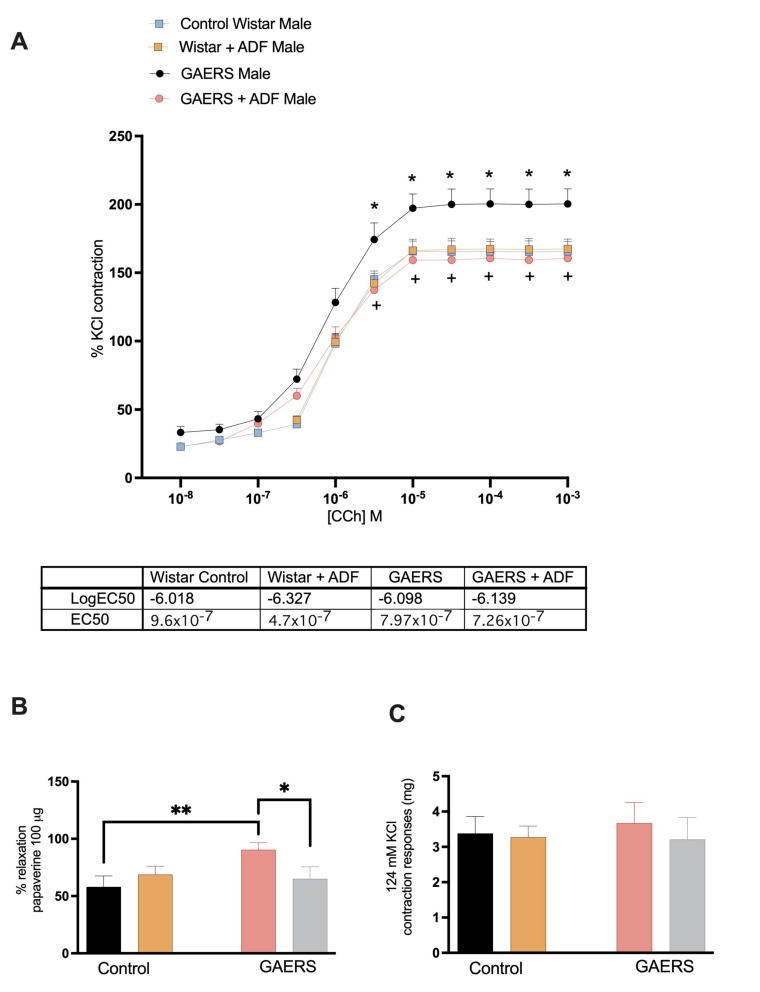
The bladder function data results from in vitro bladder contraction–relaxation responses. (**A**) Bladder contraction responses to CCh, (**B**) relaxation responses to papaverine, and (**C**) contraction responses to KCl in experimental groups. (*: *p* < 0.05, **: *p* < 0.01).

**Figure 2 biology-14-00158-f002:**
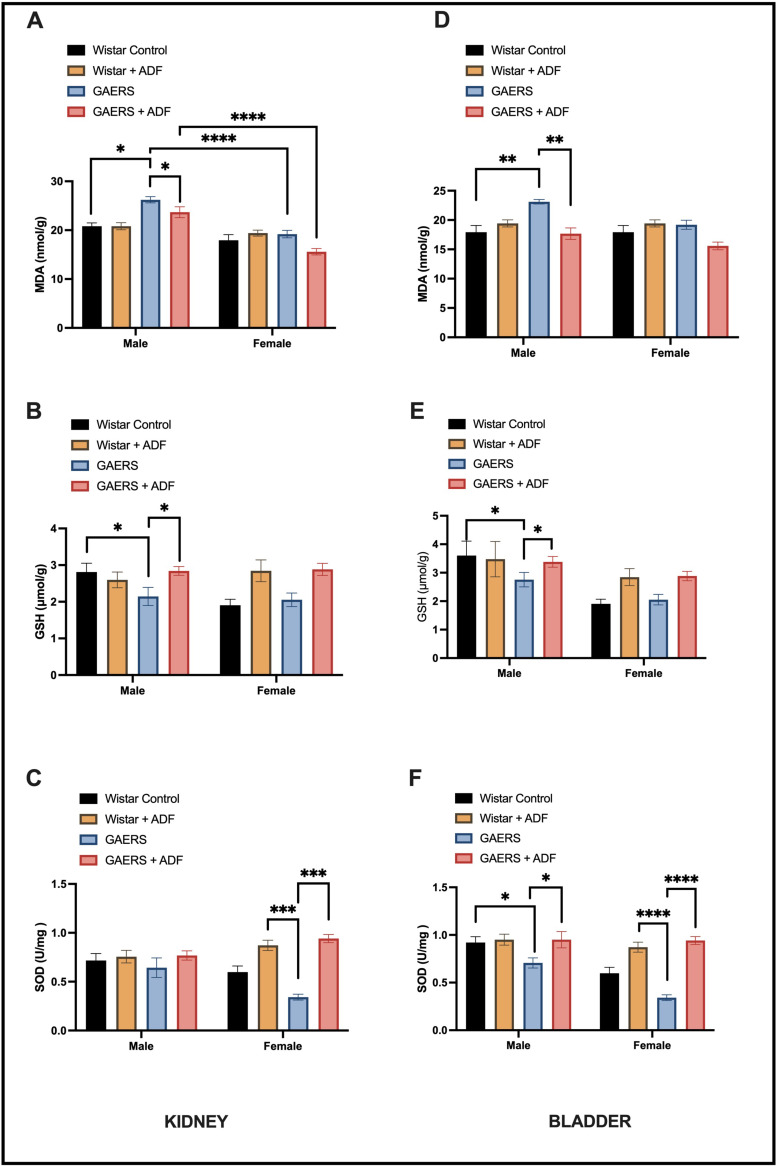
Differences in the oxidative stress markers (**A**,**D**) MDA, (**B**,**E**) GSH and (**C**,**F**) SOD for the kidney and bladder tissues between the experimental groups (*: *p* < 0.05, **: *p* < 0.01, ***: *p* < 0.001, ****: *p* < 0.0001).

**Figure 3 biology-14-00158-f003:**
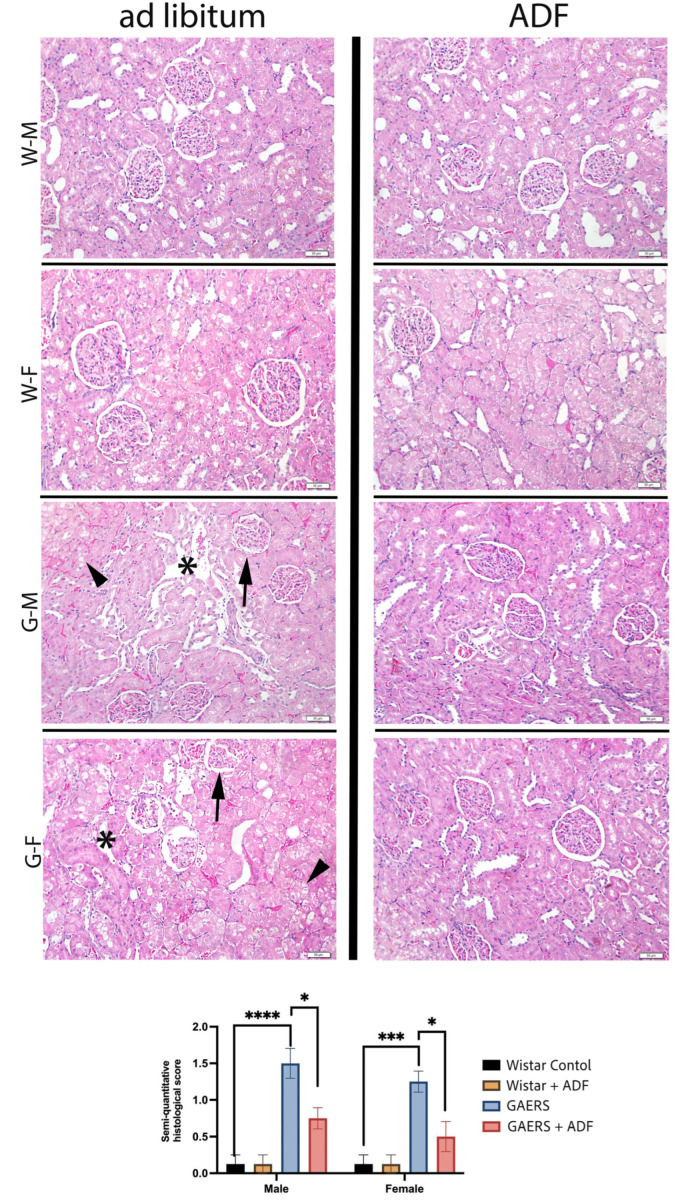
Representative micrographs of kidney tissues from all experimental groups and the graphics of semi-quantitative histopathological scoring statistical analyses. Arrowhead: tubular degeneration, arrow: damaged glomerulus with dilated Bowman’s space, asterisk: dilated tubules. WF: Wistar female. WM: Wistar male. GF: GAERS female. GM: GAERS male. Ad libitum: normal diet. IF: intermittent fasting. Hematoxylin and eosin staining. Bars: 50 µm (*: *p* < 0.05, ***: *p* < 0.001, ****: *p* < 0.0001).

**Figure 4 biology-14-00158-f004:**
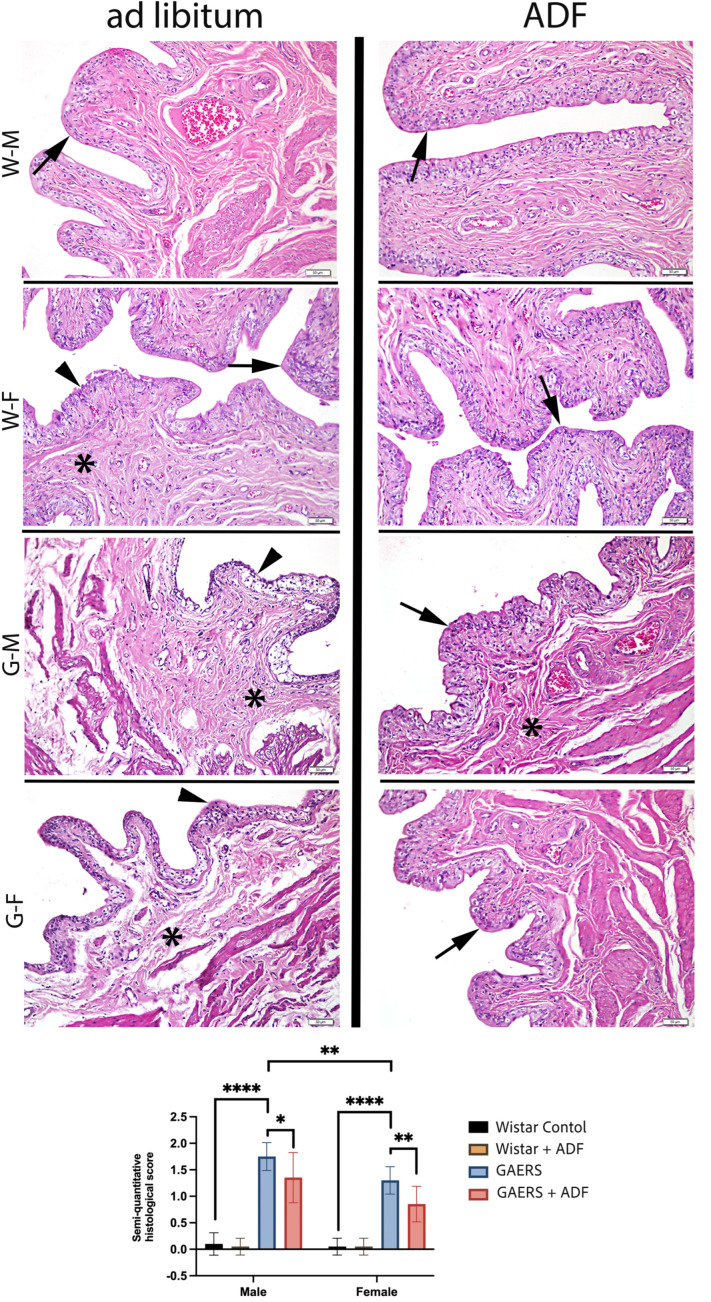
Representative micrographs of bladder tissues from all experimental groups and the graphics of statistical analyses of semi-quantitative histopathological scoring. Arrow: urothelium with regular appearance, arrowhead: irregular urothelium, asterisk: edema in connective tissue and thin muscle bundles. WF: Wistar female. WM: Wistar male. GF: GAERS female. GM: GAERS male. Ad libitum: normal diet. ADF: alternative day fasting. Hematoxylin and eosin staining. Bars: 50 µm (*: *p* < 0.05, **: *p* < 0.01, ****: *p* < 0.0001).

**Figure 5 biology-14-00158-f005:**
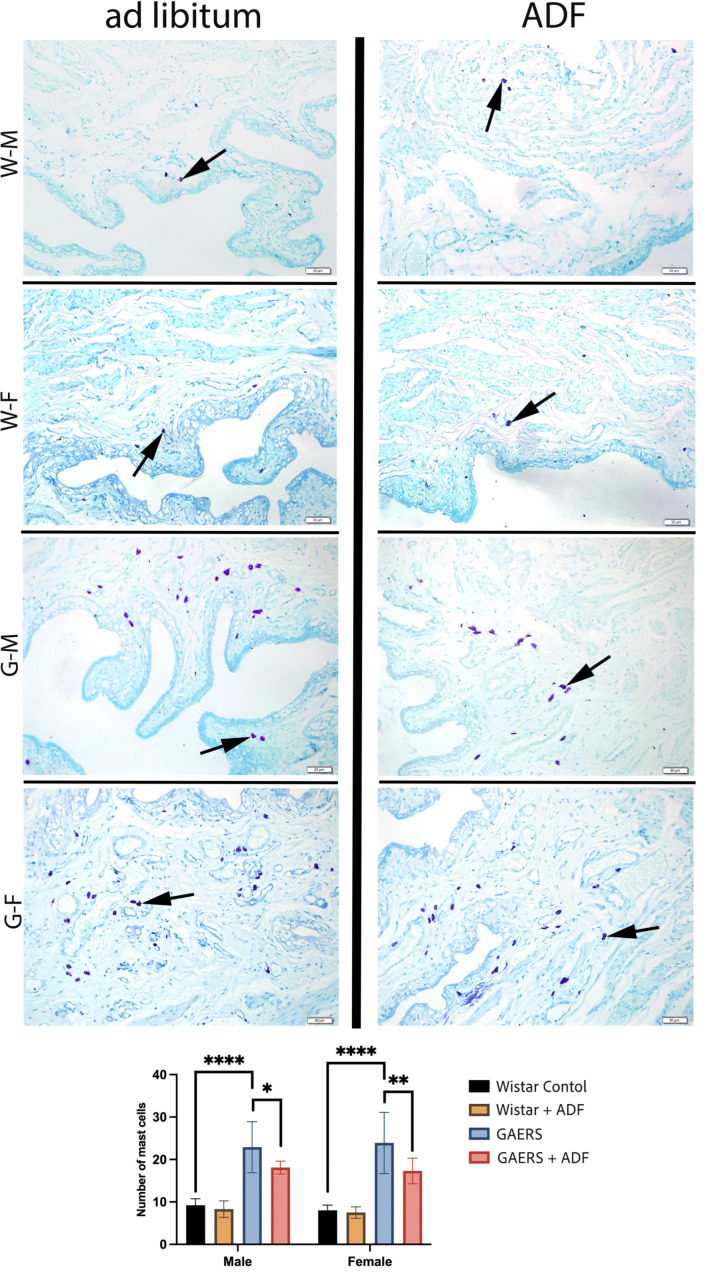
Representative micrographs of bladder tissues from all experimental groups and the graphics of statistical analyses of the mast cell counting. Arrow: mast cell. WF: Wistar female. WM: Wistar male. GF: GAERS female. GM: GAERS male. Ad libitum: normal diet. ADF: alternative day fasting. Toluidine blue staining. Bars: 50 µm (*: *p* < 0.05, **: *p* < 0.01, ****: *p* < 0.0001).

## Data Availability

The data supporting the findings of this study are available in this paper. All the obtained data are shared in the main text and Appendix A.

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
