# Peer review of "The Sex-Dependent Ameliorative Effect of Intermittent Fasting on Urinary System Functions in Genetic Absence Epileptic Rats†"

_biology, 2025, doi:10.3390/biology14020158_

Round 1
Reviewer 1 Report
Comments and Suggestions for Authors
Thank you for the opportunity to review the intriguing manuscript titled: "The Gender-Dependent Ameliorative Effect of Intermittent Fasting on Urinary System Functions in Genetic Absence Epileptic Rats." This study presents novel insights and contributes significantly to the existing body of knowledge regarding the impact of IF/ADF on renal health and function.
However, the authors need to integrate their findings with the current literature on IF. This includes contextualizing their results within the broader framework of existing research on the health benefits and disease-preventive effects associated with IF regimens. By doing so, the authors can enhance the relevance and applicability of their study, providing a more comprehensive understanding of the implications of their findings in both clinical and research settings. Attached is the annotated PDF with more than 50 comments and edits suggested.

Author Response
Comments 1: Change gender to sex. The term gender is typically more associated with social and cultural roles and behaviors in humans. In scientific contexts, especially when discussing biological and physiological differences, the term "sex" is used. |
Response 1: Thank you for this important suggestion. We completely agree with the distinction between “gender” and “sex” in scientific contexts. In line with your recommendation, we have replaced all instances of the term “gender” with “sex” throughout the manuscript to ensure consistency and accuracy when referring to biological and physiological differences. The changes have been highlighted in yellow in the revised version for easy reference. We would also like to thank the referee for guiding us to use the correct wording to make the article truly sex-focused, considering the hypothetical differences between sex and gender. |
Comments 2: The word "studies" dictates citing more than one reference. Add reference to this paragraph. |
Response 2: Thank you for the correction, we added the references in line 40 and labelled in yellow. |
Comments 3: Suggestions about the abbreviation usage. |
Response 3: Thank you for the comment. All abbreviations in the draft text were reviewed in line with your recommendations. The full form of the abbreviation followed by the abbreviation in parentheses was provided at its first mention, and only the abbreviation was used in subsequent sections labelled in yellow. |
Comments 4: More recent and relevant study could be used: https://www.mdpi.com/1648-9144/60/1/191. |
Response 4: Thank you for highlighting this. We added the suggested reference to the introduction in line 63. |
Comments 5: “These dietary factors also significantly impact inflammation linked to central nervous system (CNS) neurodegenerative disorders” Need rephrasing and correction. |
Response 5: Agree. We have, accordingly, modified the expression as follow “These dietary factors play a significant role in modulating inflammation associated with central nervous system (CNS) neurodegenerative disorders, such as epilepsy, Alzheimer’s disease, Parkinson’s disease, normal aging, amyotrophic lateral sclerosis, and multiple sclerosis.” in line 74-77. |
Comments 6: Before delving into seizure and inflammation/oxidative stress, it would be more logical to mention the anti-inflammatory and anti-oxdint effects of IF models such as Ramadan IF model: https://www.nature.com/articles/s41598-023-43862-9 https://www.sciencedirect.com/science/article/pii/S2352385918300744 https://www.sciencedirect.com/science/article/abs/pii/S0271531712001820?via%3Dihub |
Response 6: Thank you for highlighting this. The suggested reference was added to the introduction section line 79-81. |
Comments 7: Be more specific, ADF |
Response 7: Agree. We have, accordingly, modified all the expression of IF as ADF in the text and figures. |
Comments 8: mean 20/20, M/F? |
Response 8: Thank you for highlighting this. We clarified the number of animals used in the experimental procedures in the line 110-111. |
Comments 9: Cite the reference method followed in this test. |
Response 9: Thank you for pointing this out. We have now included the appropriate reference to the methodology used in this test, as suggested. |
Comments 10: Be consistent. It is important to know that even in human research, gender and sex can not be used interchangeably, as they refer to distinct concepts. "Sex" pertains to the biological differences between males and females, including genetic, anatomical, and physiological variations. In contrast, "gender" encompasses the roles, behaviors, activities, expectations, and societal norms that cultures and societies attribute to individuals based on their sex; it is often regarded as a social and cultural construct. Therefore, using these terms accurately is crucial for clarity in research, allowing for more precise discussions about biological factors versus social and cultural influences. |
Response 10: Thank you for this insightful comment. We completely agree with the importance of distinguishing between “sex” and “gender” to maintain accuracy and clarity in scientific research. As you highlighted, “sex” refers to biological differences, while “gender” pertains to social and cultural constructs. In light of this, we have reviewed the manuscript thoroughly and ensured that “sex” is consistently used when referring to biological differences. These changes have been implemented throughout the text and highlighted in the revised version for clarity. We appreciate your attention to this critical distinction, which enhances the precision and scientific integrity of our manuscript. |
Comments 11: "In vitro" does not need to be italicized in scientific writing, as it is a common term that is widely accepted in this form. https://blog.mdpi.com/2022/01/17/common-english-errors/#:~:text=Here's%20a%20short%20list%20of,%E2%80%9D%20or%20%E2%80%9Cex%20situ%E2%80%9D. |
Response 11: Thank you for bringing this to our attention. We agree with your observation that “in vitro” does not need to be italicized in scientific writing, as it is widely accepted in its standard form. Following your suggestion, we have removed the italics from all occurrences of “in vitro” throughout the manuscript ie. line 208. These changes have been marked in the revised version for your convenience. |
Comments 12: Add as supplementary data & Why only male rodents were used here? please elaborate. This should be clear in the figure caption. Also, replace IF with ADF, as there are many types and models of IF. This should be applied to the rest of figures and text. |
Response 12: To make it clear CCh results of female rats were given in supplementary data. |
Comments 13: Make it: Differences in the oxidative stress markers for the kidney and bladder tissues between the experimental groups. |
Response 13: Dear referee, thank you for pointing out the comparisons we had inadvertently overlooked. The comparisons in females in antioxidant parameters (bladder and kidney) were marked and the comparisons were stated in the results section. |
Comments 14: applicaiotn |
Response 14: Thank you for highlighting this. It is corrected in the line 272. |
Comments 15: The phrase ad libitum is not used in the text, except in the figure legends. The phrase has to be used in the text for more consistency between text and figure. |
Response 15: Thank you for pointing out this inconsistency. We have revised the manuscript to include the phrase ad libitum in the main text where appropriate, ensuring consistency between the text and the figure legends. These additions can be found in the revised version in the line 119-122, with changes highlighted for clarity. |
Comments 16: This could be supported by previous research on IF, using similar and other anti-oxidant parameters: https://www.frontiersin.org/journals/nutrition/articles/10.3389/fnut.2024.1437169/full https://onlinelibrary.wiley.com/doi/10.1155/2012/802924 One meta-analysis on RIF showed ameliorating effect on MDA: https://www.sciencedirect.com/science/article/pii/S2352385918300744 https://link.springer.com/article/10.1007/s11011-024-01415-7 |
Response 16: Thank you for providing these valuable references. We have reviewed the suggested studies and incorporated relevant findings into the manuscript to support our discussion of intermittent fasting (IF) and its effects on antioxidant parameters, including MDA. These additions help contextualize our results within the broader framework of existing research. The references have been added to the discussion section page 14, with appropriate citations, to strengthen the manuscript. The changes are highlighted in the revised version for clarity. |
Comments 17: Changes observed in the SOD and other anti-oxidant and anti-inflammatory parameters could be explained at the genetic level by virtue of the effect of IF on gene expressions: https://www.sciencedirect.com/science/article/abs/pii/S0168822719302177 |
Response 17: Thank you for highlighting this important aspect and for providing the relevant reference. We have incorporated the suggested study into the discussion section to explore the potential genetic mechanisms underlying the effects of intermittent fasting (IF) on SOD and other antioxidant and anti-inflammatory parameters. This addition helps explain the observed changes at the molecular level, specifically in relation to the influence of IF on gene expression. The reference has been cited appropriately in the line 433, and the discussion has been updated accordingly. These revisions are highlighted in the revised version for your convenience. |
Comments 18: Could be supported by other relevant research: https://www.sciencedirect.com/science/article/abs/pii/S0271531712001820?via%3Dihub https://pubmed.ncbi.nlm.nih.gov/37833312/#:~:text=Ramadan%20intermittent%20fasting%20is%20associated,subjects%20with%20obesity%3A%20lipidomics%20analysis |
Response 18: Thank you for the valuable suggestion. The some of the other suggested references has been cited appropriately in the line 433, and the discussion has been updated accordingly. These revisions are highlighted in the revised version for your convenience. |
4. Response to Comments on the Quality of English Language |
Point 1: The text of the manuscript has a writing problem. Please have it reviewed and corrected by a native expert. |
Response 1: Thank you for highlighting this. The manuscript has undergone thorough proofreading by a native speaker to ensure grammatical accuracy and fluency. The current version reflects these corrections and improvements. |

Reviewer 2 Report
Comments and Suggestions for Authors
Dear Authors
I read your manuscript carefully; your work is acceptable with minor revise. Please apply the following comments.
Article entitled “The Gender-Dependent Ameliorative Effect of Intermittent Fasting on Urinary System Functions in Genetic Absence Epileptic Rats” has been written in a good way; It is an interesting topic. Your work is acceptable with minor revision. Please apply the following comments.
-I could not see the Abstract section in pdf version of manuscript.
-Your keywords should be write based on MeSH terms.
Introduction:
In the introduction, you should first discuss the role of complementary therapies (natural compounds and traditional medicine and also intermittent fasting) on inflammation and oxidative stress in health and chronic disease especially its effects on urinary tract. You can also use the following references in your article for strengthen the introduction and discussion sections. These references are helpful.
1) Hassan, H. M., et al. (2024). Potential role for vitamin D vs. intermittent fasting in controlling aquaporin-1 and aquaporin-3 expression in HFD-induced urinary bladder alterations in rats. Frontiers in Molecular Biosciences, 10, 1306523.
2) The effect of conjugated linoleic acids on inflammation, oxidative stress, body composition and physical performance: a comprehensive review of putative molecular mechanisms (Nutrition & Metabolism).
3) Nattagh‐Eshtivani et al. The role of Pycnogenol in the control of inflammation and oxidative stress in chronic diseases: Molecular aspects.
Gamal El-Tahawy, N. F., & Ahmed Rifaai, R. (2023). Intermittent Fasting Protects Against Age-Induced Rat Benign Prostatic Hyperplasia via Preservation of Prostatic Histomorphology, Modification of Oxidative Stress, and Beclin-1/P62 Pathway. Microscopy and Microanalysis, 29(3), 1267-1276.
Methods:
-Please mention your study timeline in study design sub-section.
Results:
- OK
Discussion:
-Please complete your discussion using the references mentioned with emphasis of molecular mechanism effects of intermittent fasting.
- The text of the manuscript has a writing problem. Please be reviewed and corrected by a native expert.
Best Regard
Comments on the Quality of English LanguageThe text of the manuscript has a writing problem. Please be reviewed and corrected by a native expert.
Author Response
3. Point-by-point response to Comments and Suggestions for Authors Comments 1: I could not see the Abstract section in the pdf version of manuscript. Response 1: Thank you for the warning. There is no space for the Abstract section in the template which is the format of the journal. You can find the Abstract below:
Abstract: Epilepsy, a neurological condition marked by recurrent seizures, often leads to secondary complications affecting various body systems, including the urinary system. This study investigates the ameliorative potential of alternate day fasting (ADF) as a commonly used type of intermittent fasting (IF) on urinary system dysfunction in Genetic Absence Epilepsy Rats from Strasbourg (GAERS) and Wistar controls, focusing on gender differences. A comprehensive approach combining histological, biochemical, and physiological analyses was employed. Bladder tissues from GAERS exhibited significant hyperreactivity and impaired contractile responses compared to controls, which were improved by ADF. Biochemical assessments revealed elevated oxidative stress markers, such as malondialdehyde (MDA) and reduced antioxidant defense (glutathione [GSH] and superoxide dismutase [SOD]) in GAERS. ADF application significantly decreased MDA levels and enhanced GSH and SOD activities, both males and females. Histological examination demonstrated pronounced urothelial and tubular damage in GAERS, accompanied by inflammation marked by increased mast cell counts. ADF mitigated these pathological features, indicating its protective effects against neuroinflammation and oxidative damage. Notably, gender-specific responses were observed, with females showing better functional recovery and males displaying more significant biochemical improvements. These results highlight ADF’s potential in reducing urinary system damage in absence epilepsy, modulating oxidative stress, and restoring bladder function, while emphasizing the importance of gender considerations. Further research is warranted to explore the mechanisms underlying these gender-specific responses and to evaluate ADF’s long-term efficacy as a therapeutic strategy in managing epilepsy-related urinary dysfunction. Keywords: fasting, urination disorders, bladder, sex difference, absence epilepsy. Comments 2: Your keyword should be write based on MeSH terms. Response 2: Thank you for that valuable suggestion. We reconsidered the keywords in terms of MeSH terms as following; fasting, urination disorders, bladder, sex difference, absence epilepsy. |
Comments 3: In the introduction, you should first discuss the role of complementary therapies (natural compounds and traditional medicine and also intermittent fasting) on inflammation and oxidative stress in health and chronic disease, especially its effects on the urinary tract. You can also use the following references in your article to strengthen the introduction and discussion sections. 1. Hassan, H. M., et al. (2024). Potential role for vitamin D vs. intermittent fasting in controlling aquaporin-1 and aquaporin-3 expression in HFD-induced urinary bladder alterations in rats. Frontiers in Molecular Biosciences, 10, 1306523. 2. The effect of conjugated linoleic acids on inflammation, oxidative stress, body composition and physical performance: a comprehensive review of putative molecular mechanisms (Nutrition & Metabolism). 3. Nattagh‐Eshtivani et al. The role of Pycnogenol in the control of inflammation and oxidative stress in chronic diseases: Molecular aspects. 4. Gamal El-Tahawy, N. F., & Ahmed Rifaai, R. (2023). Intermittent Fasting Protects Against Age-Induced Rat Benign Prostatic Hyperplasia via Preservation of Prostatic Histomorphology, Modification of Oxidative Stress, and Beclin-1/P62 Pathway. Microscopy and Microanalysis, 29(3), 1267-1276. |
Response 3: Thank you for pointing this out and for the suggested references. We have revised the introduction to include a detailed discussion on complementary therapies, including intermittent fasting, natural compounds, and traditional medicine, with an emphasis on their roles in modulating inflammation and oxidative stress in chronic diseases, particularly in the urinary system. All the suggested references have been critically analyzed and integrated where relevant. These changes can be found in the Introduction section on Page 1 and 2 with the newly added references highlighted in yellow. |
Comments 4: Please mention your study timeline in the study design subsection. |
Response 4: We agree with this comment and have added a clear description of the study timeline to the Methods section as the graphical abstract (Figure 1). This addition helps clarify the sequence of experimental procedures and ensures better understanding of the workflow. The changes are marked in yellow on Page 3. |
Comments 5: Please complete your discussion using the references mentioned with emphasis of molecular mechanism effects of intermittent fasting. |
Response 5: Thank you for this insightful suggestion. We have reorganized and expanded the discussion section to emphasize the molecular mechanisms underlying the effects of intermittent fasting, focusing on neuroinflammatory pathways, oxidative stress modulation, and the potential interactions with the urinary system. The references related to suggested topic were incorporated to strengthen the discussion, and the revisions are labeled in yellow on Pages 13 and 16. |
Comments 6: The text of the manuscript has a writing problem. Please have it reviewed and corrected by a native expert. |
Response 6: Thank you for highlighting this. The manuscript has undergone thorough proofreading by a native speaker to ensure grammatical accuracy and fluency. The current version reflects these corrections and improvements. |
Comments 7: |
Response 7: Agree. I/We have, accordingly, done/revised/changed/modified…..to emphasize this point. Discuss the changes made, providing the necessary explanation/clarification. Mention exactly where in the revised manuscript this change can be found – page number, paragraph, and line.] “[updated text in the manuscript if necessary]” |
4. Response to Comments on the Quality of English Language |
Point 1: The text of the manuscript has a writing problem. Please have it reviewed and corrected by a native expert. |
Response 1: Thank you for highlighting this. The manuscript has undergone thorough proofreading by a native speaker to ensure grammatical accuracy and fluency. The current version reflects these corrections and improvements. |

Reviewer 3 Report
Comments and Suggestions for Authors
General comment
The manuscript entitled “The Gender-Dependent Ameliorative Effect of Intermittent Fasting on Urinary System Functions in Genetic Absence Epileptic Rats” explores a novel area of research, examining the impact of intermittent fasting (IF) on urinary system functions in a rodent model of absence epilepsy. The findings are relevant to the field of neurophysiology and urology, especially regarding gender-specific therapeutic approaches. However, there are areas where the manuscript could be strengthened for better clarity and scientific rigor. In detail, issue requiring revision are:
- While the study demonstrates significant findings, the discussion on mechanisms linking epilepsy, inflammation, and urinary dysfunction could be expanded.
- The rationale for the number of animals in each group should be included.
- The manuscript attributes some observed gender differences to estrogen without providing sufficient evidence.
- It is unclear if appropriate controls (e.g., non-fasting epileptic animals with sham interventions) were used to rule out other factors influencing the results.
Author Response
Please see the attachment.
Comments 1: While the study demonstrates significant findings, the discussion on mechanisms linking epilepsy, inflammation, and urinary dysfunction could be expanded. |
Response 1: Thank you for highlighting this. We have expanded the discussion to include a more detailed exploration of the molecular mechanisms linking epilepsy, inflammation, and urinary dysfunction. Recent studies on neuroinflammatory pathways, cytokine involvement, and oxidative stress markers have been incorporated to provide a comprehensive analysis. Please see the yellow labelled sections in the introduction and discussion of the revised manuscript. |
Comments 2: The rationale for the number of animals in each group should be included. |
Response 2: We appreciate this observation. The rationale for the sample size was determined based on previous studies using similar experimental designs and statistical power calculations. We have included this explanation in the Methods section under the Animals subsection. Please see Page 3, line 109-110. |
Comments 3: The manuscript attributes some observed gender differences to estrogen without providing sufficient evidence. |
Response 3: Thank you for this important point. We have revised the text to avoid overgeneralizations and added a more cautious interpretation of the observed sex differences. While estrogen’s role is discussed as a potential contributing factor, additional factors, such as hormonal fluctuations and sex-specific responses to neuroinflammation, are now acknowledged. Relevant citations have been added to support this balanced perspective. Please see the yellow labelled lines in the introduction and discussion parts. |
Comments 4: It is unclear if appropriate controls (e.g., non-fasting epileptic animals with sham interventions) were used to rule out other factors influencing the results. |
Response 4: Thank you for this observation. We have clarified the inclusion of appropriate control groups in the Methods section line 107-109. Specifically, we included non-fasting epileptic animals and standard diet (ad libitum) groups to distinguish the effects of intermittent fasting from other potential confounders (non-fasting animals as shamcontrol groups for the fasting). |

Round 2
Reviewer 3 Report
Comments and Suggestions for Authors
The authors improved the manuscript according to previous suggestions. No further corrections required.
Author Response
Comment: Please keep the group names consistent in all figures. When you use "Wistar control," you should not use "Control Wistar" in Figure 1A. When you use "GEAR+ADF," you should not use "ADF + GEAR" in Figures 4 and 5.
It is unclear which groups had ADF in Figures 1B and 1C because the colors in Figure 1A did not wholly match their colors.
Supplementary data should be checked for grammar as well. Table 1 was not mentioned in the results of this manuscript.
Response: Thank you for your valuable comments and suggestions, which have greatly contributed to improving the quality of our manuscript. We have carefully addressed the points raised and made the necessary revisions:
-
Group Name Consistency: The group names in Figure 1A and Figures 4 and 5 have been updated to ensure consistency throughout the manuscript. Specifically, we have standardized the naming to "Wistar Control" and "GAERS + ADF" in all figures.
-
Figure 1 Adjustments: Figure 1 has been revised to align with the suggestions, ensuring that the colors in Figures 1B and 1C correspond accurately to those in Figure 1A.
-
Supplementary Data: The grammar in the supplementary data has been reviewed and corrected. Additionally, an updated supplementary file has been included in the revised submission.
-
Table 1 Reference: Table 1 has been properly referenced and incorporated into the results section of the manuscript as suggested.
We appreciate your careful review and insightful feedback, which have helped us improve the clarity and presentation of our work.